# Effect of Homogenization Modified Rice Protein on the Pasting Properties of Rice Starch

**DOI:** 10.3390/foods11111601

**Published:** 2022-05-29

**Authors:** Jianyong Wu, Shunqian Xu, Xiaoyan Yan, Xuan Zhang, Xingfeng Xu, Qian Li, Jiangping Ye, Chengmei Liu

**Affiliations:** 1State Key Laboratory of Food Science and Technology, Nanchang University, Nanchang 330047, China; jianyongwu@ncu.edu.cn (J.W.); xushunqian@aliyun.com (S.X.); yanxiaoyan2021@email.ncu.edu.cn (X.Y.); bqsehj6ara@163.com (X.Z.); liuchengmei@ncu.edu.cn (C.L.); 2Department of Food Science and Engineering, Qingdao Agricultural University, Qingdao 266109, China; jiaonanxuxingfeng@126.com; 3Key Laboratory of Fermentation Engineering (Ministry of Education), Hubei University of Technology, Wuhan 430068, China; lqh198299@126.com

**Keywords:** rice protein, homogenization, rice starch, gelatinization, rheology

## Abstract

Modification of plant-based protein for promoting wide applications is of interest to the food industry. Rice protein from rice residues was modified by homogenization, and its effect on pasting properties (including gelatinization and rheology) of rice starch was investigated. The results showed that homogenization could significantly decrease the particle size of rice protein and increase their water holding capacity without changing their band distribution in SDS-PAGE. With the addition of protein/homogenized proteins into rice starch decreased peak viscosity of paste. The homogenized proteins decreased breakdown and setback value when compared with that of original protein, indicating homogenized protein might have potential applications for increasing the stability and inhibiting short-term retrogradation of starch paste. The addition of protein/homogenized proteins resulted in a reduction in the viscoelasticity behavior of starch paste. These results indicate that homogenization would create a solution to alter the physicochemical properties of plant proteins, and the homogenized proteins may be a potential candidate for development of protein-rich starchy products.

## 1. Introduction

Rice protein, a high-quality vegetable protein, is one of the main ingredients of rice. Rice protein has attracted wide attention because of its hypoallergenic, rational amino acid composition, lower cholesterol and high nutritional value [1]. In addition, rice protein (including rice protein hydrolysate and its specific peptide components) has antioxidant, anti-hypertensive, anti-cancer and anti-obesity effects [2]. Native rice protein contains four components: albumin, globulin, gliadin and glutelin, among which glutelin content is relatively high, accounting for 66–78% of the total rice protein [3]. However, the peptide molecules in glutelin can be cross-linked with each other through disulfide bonds and hydrogen bonds to form large and complex aggregates, so that the rice protein is more hydrophobic and less soluble [4], thereby greatly restricting the application of rice protein. Consequently, there is considerable interest in improving the functional properties of rice protein that are easily applied to food products.

Currently, efforts have been made to modify proteins using a variety of methods, including physical methods such as heating [5], extrusion [6], pulsed electric field [7], static high pressure [8], micro-jet [9] and ultrasonic [10]; chemical methods such as deamidation [11], phosphorylation, glycation and acylation [12,13]; and enzyme modification methods such as crosslinking [8] and hydrolysis [14]. These processing technologies may effectively affect solubility, molecular weight, water-holding capacity and surface hydrophobicity properties of proteins, resulting in a profound impact on the properties of starch-based food products [15].

Previous studies have shown that higher soluble protein had greater influence on the gelatinization properties of starch [16]. It is also indicated that small molecular weight proteins had a stronger effect on the inhibition of starch retrogradation or digestibility than large molecular weight proteins [17,18]. Therefore, it is feasible to regulate the physico-chemical properties of starch-based food products by changing the properties of protein. Studying the impact of modified proteins on starch properties may be helpful to interpret some phenomenon of proteins in starchy food, and broaden utilization of proteins in the food industry, as well as offer other possibilities for constructing protein-enriched foodstuff (such as cereal beverage, baby rice cereal).

Homogenization is a non-thermal physical processing technology that can impact protein aggregation and conformation via cleavage of noncovalent bonds. It forces the protein slurry to pass through a narrow channel at a high velocity, during which the protein aggregates would be unfolded and dissociated by high-frequency vibration, instantaneous pressure drop, intense shear and cavitation. Up until now, homogenization was used to modify many kinds of proteins such as soybean protein [19], faba bean protein [20], lentil protein [21] and whey protein [22], etc. However, the feasibility of this technique on rice protein and the further application of resultant proteins were still needed. Therefore, the objective of the present paper was to investigate the effects of homogenization on physico-chemical properties of rice proteins, and the effect of these modified rice proteins on the pasting properties of rice starch was then investigated with the expectation of developing protein-enriched starchy food.

## 2. Materials and Methods

### 2.1. Materials

Rice protein (93.4 wt%, dry basis) and rice starch (with 8.0% moisture content, 22.4% amylose content) were provided by Jiangxi Jinnong Biotechnology Co., Ltd. (Shanggao, China). Bradford protein concentration determination kit and sodium dodecyl sulfate polyacrylamide gel electrophoresis kit were purchased from Solarbio Biotechnology Co., Ltd. (Beijing, China). Chromatographically pure potassium bromide was purchased from Sigma-Aldrich (St. Louis, MO, USA). All other chemical reagents were of analytical grade.

### 2.2. Homogenization of Rice Protein and Sample Preparation

A suspension of rice protein with distilled water 1:10 (*w:w*) was homogenized with a homogenizer (SRH50-80, Shanghai SAMRO homogenizer, Shanghai, China) at a pressure of 30 MPa for 0, 10, 20 and 30 min. Suspensions were then centrifuged at 4000× *g* for 5 min and the precipitate was freeze-dried. The corresponding samples obtained from this process were labeled as P0, P10, P20 and P30, where the numbers represented the treatment time during the homogenization process. The treated rice protein was then mixed with rice starch at a ratio of 1:9 (dry basis, *w*/*w*) for further analysis, marked as S-P0, S-P10, S-P20 and S-P30, respectively. The rice starch alone (S), without rice protein, was used as a control sample.

### 2.3. Characterization of Protein Properties

#### 2.3.1. Particle Size Analysis

The particle size of proteins or modified proteins was determined by laser diffractometer (Mastersizer 3000, Malvern Instruments Ltd., Worcester, UK) based on dynamic light scattering [23]. Samples were diluted into a 1% suspension with distilled water. The dispersion was vortexed for 2 min, and then the particle size measurement was carried out immediately. The refractive indexes of water and protein were 1.33 and 1.52, respectively, and the shading degree was set to 5–8%. The particle size distribution and specific surface area of rice protein and modified proteins were obtained through testing.

#### 2.3.2. Gel Electrophoresis Analysis

Sodium dodecyl sulfate polyacrylamide gel electrophoresis (SDS-PAGE) was carried out on a discontinuous buffer system according to the method of previous study [24]. Samples were dissolved in 0.3% sodium hydroxide solution and dispersed in loading buffer containing 0.5 M Tris-HCl, 1.0% SDS, 2.0% mercaptoethanol, 0.05% bromophenol blue and 5% glycerin, then heated in boiling water for 5 min. After cooling and centrifugation, 10 μL of supernatant was loaded on the 4% stacking gel and 12% separating gel, with an electric current in the stacking gel of 8 mA and an electric current after entering the separation gel of 16 mA. When electrophoresis was over, the gels were stained with the staining solution of coomassie brilliant blue R250, and destaining with methanol-glacial acetic acid solution until the background was near colorless. The molecular weight markers used were 11, 17, 20, 25, 35, 48, 65, 75, 100 and 155 kDa.

#### 2.3.3. Water Holding Capacity Analysis

The water holding capacity (WHC) was measured according to the method reported previously [25]. Briefly, WHC was determined by weighting about 1.0 g protein samples into 10 mL distilled water, and the suspension was kept at room temperature for 24 h. After that, the supernatant was removed by suction filtration, and the hydrated residue weight was obtained after draining the dripping water. Then, the residual was dried at 105 °C for 12 h to obtain its dry weight. WHC could be calculated by the ratio of the water content in protein and protein weight.

#### 2.3.4. Microstructure Analysis

The protein samples (P0, P10, P20 and P30) were dispersed in distilled water (10.0% wt) before being placed on a glass slide and covered with a thin glass clover slip. The microstructure of the samples was observed by an inverted optical microscope (CKX41, Olympus, Japan), with a 40× objective and 10× eyepiece, and pictured by Nikon image analysis software.

### 2.4. Characterization of Pasting Properties of Protein-Starches

#### 2.4.1. Rapid Visco-Analyzer Analysis of Protein-Starch Paste

Pasting properties of starch-protein paste were carried out by rapid visco-analyzer (RVA, Perten Instruments, Australia) as described previously [26]. The mixed samples, as shown in Section 2.2, were directly weighted (3.0 g) into the RVA canister and 25.0 g of distilled water was added. The measurement was then carried out according to an inherent program that came with instruments as *rice rapid*. Briefly, the sample was initially held at 50 °C for 1 min, then heated to 95 °C with the rate of 12 °C/min, and held at 95 °C for 2.5 min. It was subsequently cooled to 50 °C at a constant rate of 6 °C/min and was held at 50 °C for 1.4 min. Agitation speed was fixed at 960 rpm for the first 10 s to ensure the uniformity of the dispersion, and then at 160 rpm throughout the rest of the measurement. The characteristic parameters of paste were obtained from the RVA curve, including the peak viscosity, trough viscosity, final viscosity, breakdown value, setback value and pasting temperature of the sample.

#### 2.4.2. Rheology of Protein-Starch Paste

The flow curve and flow behavior of the protein-starch were determined using a rheometer (MCR302, Anton Paar, Austria) with a parallel-plate geometry of diameter 50 mm and measuring gap of 1 mm. The mixed sample suspension (250 mg samples + 1.0 mL H_2_O) was conditioned at 25 °C for 1 min. The gap edge was then covered with a thin layer of low-density silicone oil (with 50 mPa viscosity) to minimize water evaporation.

For steady shear test analysis, at a shear stress of 5 Pa, the suspension was heated from 25 to 95 °C, and then cooled from 95 to 25 °C, with a rate of 5 °C/min. After that, the sample was sheared from 0.1 to 200 s^−1^ at 25 °C, and the changes of shear stress and viscosity values with shear rate were recorded. The data were fitted using a power law model as follows:$$\sigma =K{\dot{\gamma }}^{n}$$

Here, σ was the shear stress (Pa), *K* was the consistency coefficient (Pa·s^n^), $\dot{\gamma }$ was the shear rate (s^−1^), n was the flow behavior index.

For dynamic oscillatory analysis, the forming paste process was the same as described above. The resulting gel was equilibrated at 25 °C for 30 s before a frequency sweep was conducted over the range of 0.1 to 20 Hz at 1% strain (within the linear viscoelastic region) and 10 points per decade at 25 °C, to characterize the viscoelastic properties of protein-starch pastes. The mechanical spectra were recorded as a function of angular frequency (Hz), including storage modulus (G′), and loss modulus (G″), and the loss tangent (tan δ = G″/G′).

#### 2.4.3. Microstructure of Protein-Starch Paste

Scanning electron microscopy (SEM) was used to observe the microstructure of the freeze-thawed protein-starch paste. Briefly, the samples were adhered to double side tape and coated with a thin film (10 nm) of gold using a vacuum evaporator. Then, the samples were observed on a SEM (Quanta-200, FEI Inc., Eindhoven, The Netherlands) with an accelerating voltage of 20 kV at 500× magnifications.

### 2.5. Statistical Analysis

All experiments were repeated at least three times to obtain the mean and standard deviation. The results were expressed as mean ± standard deviation. The statistical analyses were performed with SPSS Statistics software (version 25, SPSS Inc., Chicago, IL, USA). A comparison of the means was performed with Tukey’s Post-hoc test at 5% level of significance using a one-way analysis of variance (ANOVA).

## 3. Results and Discussion

### 3.1. Effect of Homogenization on Physicochemical Properties of Protein

#### 3.1.1. Effect of Homogenization on Average Particle Size of Rice Protein

The mean particle size of homogenization treated rice protein under different times is shown in Table 1. The mean particle size D50 of native rice protein was 24.2 ± 0.3 μm, which was about the same as the size in the previous study [27]. It was found that homogenization time had an obvious influence on particle size (*p <* 0.05). After the homogenization treatment, the mean particle size of P10, P20 and P30 was decreased to 13.8 ± 0.2 μm, 10.8 ± 0.1 μm and 8.74 ± 0.4 μm, respectively. This phenomenon might be due to the fact that mechanical forces generated by homogenization could disrupt the structure of protein molecules, resulting in the reduction in particle size. Similarly, previous studies reported that homogenization reduced particle size of lentil protein [21] and faba bean protein [20].

#### 3.1.2. Effect of Homogenization on Protein Profiles of Rice Protein

The electrophoretic profiles of the rice protein that had been homogenization treated for different times are shown in Figure 1. Rice protein had two main bands at about 11 kDa and 11–17 kDa, and shallow bands at 17–20 kDa, about 35 kDa, about 63 kDa and above 245 kDa. According to the previous study, the two main bands at about 11 kDa and 11~17 kDa might belong to prolamine and globulin [28]. However, the content of prolamine in rice protein was very low. Thus, the band at about 11 kDa in rice protein was considered to the glutelin, and the subunits at 17–20 kDa, about 35 kDa and about 63 kDa belong to β-gluten, α-gluten and the precursor of glutelin. The shallow band at above 245 kDa might belong to the aggregation of protein that cannot enter the gel due to their large molecular weight. As shown in Figure 1, the homogenization treatment had no significant change on the band distribution of rice protein in SDS-PAGE, which was consistent with high-pressure homogenization treated hazelnut protein isolate with increasing homogenization pressure from 0 to 150 MPa [29].

#### 3.1.3. Effect of Homogenization on WHC of Rice Protein

Water holding capacity (WHC) is an important property in the application process of rice protein. Higher water-holding capacity can reduce water loss during product processing and storage, thereby maintaining the freshness and taste of the products. The WHC value for the original rice protein sample was 1.65 ± 0.02 g/g (Table 1). After homogenization treatment, the WHC value of P10, P20 and P30 was significantly (*p* < 0.05) increased to 1.90 ± 0.03 g/g, 2.06 ± 0.04 g/g and 2.23 ± 0.08 g/g, respectively. The hydration properties of rice protein are generally considered to be related to its component content and structure. As shown above, homogenization treatment may not change the content of protein components in rice protein. Therefore, the changes in content of disulfide bonds, surface hydrophobicity and protein secondary structure were considered to be important factors affecting protein water holding capacity [1]. In addition, the increase in WHC in the present study might also be ascribed to the smaller particle sizes of homogenization treated rice proteins.

#### 3.1.4. Effect of Microstructure of Rice Protein

Microstructural images have an important implication to explain the changes of rice protein. The microstructural images of homogenization treatment on rice protein suspensions were shown in Figure 2. It was indicated that homogenization had a significant effect on the protein particles. As shown, non-homogenized rice protein suspension consisted mainly of large particles with an irregular shape and some of the particles that could be flocculated. Homogenization treatment for 10 min caused a decrease in particle size results when compared with the control sample, the number of small particles in the suspension increased with few bigger particles remaining. It is obvious that when the homogenization treatment time increased to 20 min or 30 min, the suspended particles were effectively broken into small particles. This result strongly supported the changes of particle size. Similar observations were determined from homogenization treatment on peanut proteins [30] and soy protein isolate [31].

Previous studies had shown that homogenization could effectively modify the functionalities of rice proteins, and exploring protein rich food products has been of great interests in the food industry [15]. However, relatively little information is available on the effect of homogenized protein on pasting properties of starch-based food products. Therefore, rice protein or homogenization treated rice protein was alternatively blended with rice starch to investigate the potential application of starchy foods in the present study.

### 3.2. Effect of Homogenized Rice Protein Addition on Pasting Properties of Rice Starch

#### 3.2.1. RVA Pasting Profile

The RVA (Rapid Visco-Analyzer) profiles and the resulting data of rice starch and protein/homogenized protein-rice starch mixtures are shown in Figure 3 and Table 2. It was observed that protein or homogenized proteins could decrease the peak viscosity (PV) and trough viscosity (TV), while the pasting temperature displayed an opposite trend. Pasting temperature was the temperature at which viscosity began to rise. As shown, the pasting temperature of rice starch was significantly increased with the addition of rice protein or homogenized rice proteins, which was closely related to the water binding ability of starch. The higher pasting temperature of S-P30 could be explained with its higher water absorption capacity, thereby reducing the available water for rice starch on account of competition for water in the presence of homogenized rice protein. Thus, the swelling of starch granules was inhibited. This was consistent with the previous research that rice protein increased the gelatinization temperature of rice starch [32].

As shown in Table 2, the peak viscosity, trough viscosity, breakdown viscosity, final viscosity and setback viscosity of native rice starch were 2718 cP, 2085 cP, 633 cP, 3245 cP and 1159 cP, respectively, and the gelatinization temperature was 76.4 °C, which was similar to results of previous studies [33]. The gelatinization of starch is a process where starch granules swell until the granule structure is completely disrupted, when starch is heated in a certain amount of water, which is accompanied by the destruction of crystal structure, the leaching of amylose, and the increase in viscosity of the system [34,35]. Peak viscosity (PV) is the maximum viscosity of starch developed soon after the heating portion of the test, which reflects the ability of starch granules to swell to the equilibrium point. After that, the rate that the starch granules break and disintegrate is higher than the swelling rate. As shown, the PV value was significantly (*p* < 0.05) decreased to 2047 cP, 2156 cP, 2208 cP and 2341 cP for rice starch blended with P0, P10, P20 and P30, respectively. The magnitude of PV decreasing for homogenized protein-starch was less than that of protein-starch, which might be primarily due to the smaller particle size resulting in higher interaction ability of homogenized protein and starch. Proteins can be denatured as the temperature rises during the heating process and some changes or interactions might occur between them, thereby increasing the peak viscosity of rice starch [36].

After reaching peak viscosity, the swollen starch granules are easily broken and disintegrated by stirring at the holding temperature (95 °C), causing the decrease in viscosity to trough viscosity. The viscosity reduction from peak viscosity to trough viscosity was known as breakdown (BD), which reflected the stability of the paste. The BD for S-P0 paste was 712 cP, and its value was significantly (*p* < 0.05) decreased to 580 cP, 498 cP and 418 cP for the samples of S-P10, S-P20 and S-P30, respectively. The increase in stability of cooled starch pastes was appropriate for producing foodstuff such as cereal beverages. It was indicated that protein after homogenization significantly increased the stability of paste. The more interactions and network formation with the smaller particle size increased the resistance to the mechanical shearing of the pastes from starch/protein blends [37].

The setback (ST) value measured the changes in viscosity of pastes from trough value to final viscosity when the cooling stage decreased to 50 °C, which usually indicated the short-range retrogradation of starch. The setback (ST) viscosity showed the same trends as the changes of BD viscosity. The decrease in ST value indicated the potential of homogenized proteins in delaying short-term retrogradation of starch. The addition of homogenized protein facilitates the interactions and network formation between protein-starch, and resulted in a reduction in the extent of amylose-amylose rearrangement, and thereby retarded the retrogradation [38]. The decrease of setback viscosity of starch was also observed in the addition of protein or protein hydrolysate [39]. This result indicates that homogenized proteins were good choices to inhibit short-term retrogradation of starch.

#### 3.2.2. Rheological Properties

Rheology had an insight on pasting property that was a different view from RVA. The dynamical rheological behavior including storage modulus (G′), loss modulus (G″) and loss tangent (tanδ) of native rice starch and rice starch-protein/homogenized proteins blends are shown in Figure 4A–C. The dynamic mechanical spectra of all paste samples were frequency independent over a large time scale. Additionally, the value of G′ was much larger than those of G″ and did not intersect each other throughout the frequency sweep range. It was suggested that these systems behaved as a weak solid-like gel.

The addition of protein or homogenized proteins decreased the G′ and G″ of starch gels, and the magnitude of G′ was greater than that of G″, thus indicating that the addition of protein or homogenized proteins decreased the viscoelasticity of the starch pastes. The viscoelasticity of starch pastes was controlled by the properties of starch (permanent junction zones in the network) and its interaction with addition (temporary entanglements between additives and starch, as well as loops among additives themselves in the network) [40]. The interaction of proteins/homogenized proteins with starch could be facilitated to increase the number of temporary network points, while the number of permanent network points that were represented by gel-forming junction zones of the starch decreased. Dynamic loss tangents (tan δ), a function of G″/G′, indicates the viscous and elastic character of the starch pastes [41]. The tan δ of rice starch added proteins was apparently higher than that of control samples, as shown in Figure 4C. More importantly, the higher tan δ was observed in the gels with the addition of homogenized proteins. These results indicated that homogenized proteins resulted in rice starch gel with a softer and more liquid-like structure (higher tan δ). This could be attributed to the rearrangement of rice starch being inhibited by proteins, especially the proteins with homogenization. Overall, the results of dynamic rheology might have important implications for supplementing homogenized protein that would be more suitable than protein in starchy foodstuffs (such as baby rice cereal) when liquid-like properties of starch are required.

Steady shear measurements of rice starch with protein or homogenized proteins were presented in Figure 4D, and the fitting results of the viscosity curve by the power law model (Table 2) were also studied. As shown, the apparent viscosity of all the starch paste samples decreased with the increase of shear rate. The flow behavior indexes (n) of all the samples was ranged from 0.23 to 0.27 (less than 1), suggesting that they were pseudoplastic non-Newtonian fluids. Overall, consistency coefficient (*K*) values significantly decreased after the addition of proteins, indicating that the proteins would reduce the consistency of the rice starch paste. However, the flow behavior indices (n) values had no significant difference among all the samples. As shown, the apparent viscosity and *K* value of rice starch paste had obviously decreased because of the addition of protein or homogenized proteins, which was consistent with the results of RVA. Similar results of the decrease in the consistency were shown in barley starch with the addition of proteins [37].

#### 3.2.3. Microstructure of Starch Pastes

The microstructure of the starch-protein gel with the addition of homogenized rice protein for different times observed by scanning electron microscopy was shown in Figure 2. As shown, the microstructure of the native rice starch gel displayed a spongy structure that is small and uniformly distributed. With the addition of proteins, the starch-protein gels still had a honeycomb-like network structure, which was the same found in previous studies [42]. However, clear differences of network structure were observed between rice starch gels with protein or homogenized proteins added and those without addition. With the addition of P0, the pores of the starch-protein gel network structure increased, and the size became uneven. In addition, it can be observed that the pore wall became rough with the addition of rice protein. This might be ascribed to the protein matrix, which is attached to the starch gel resulting in the rough surface. With the addition of P10, P20 and P30, the structure of the gels had many broken network structures with a much thinner fibrillar pore wall. This result indicated that the interaction between homogenized protein and gelatinized starch was involved in the formation of gel network structure, thereby reducing the formation of network structure between rice starches.

## 4. Conclusions

Overall, homogenization could effectively decrease the particle size and morphology of rice proteins without altering their primary structure along with the significant improvement in water holding capacity. The pasting viscosities of homogenization modified protein-starch mixtures were lower than those of protein-starch mixtures and native rice starch, especially with lower breakdown and setback values. The addition of homogenized proteins caused a decrease in the viscoelasticity of rice starch pastes indicating the increase in its liquid-like behavior. Therefore, homogenized rice protein may be more suitable than rice protein for increasing stability, inhibiting short term retrogradation and maintaining rheological properties of rice starch gels, which has important implications for its application in starchy products such as cereal beverage and baby rice cereals. However, further studies are required to be sure of the effect of homogenized proteins on long-term retrogradation and digestibility of starch gels.

## Figures and Tables

**Figure 1 foods-11-01601-f001:**
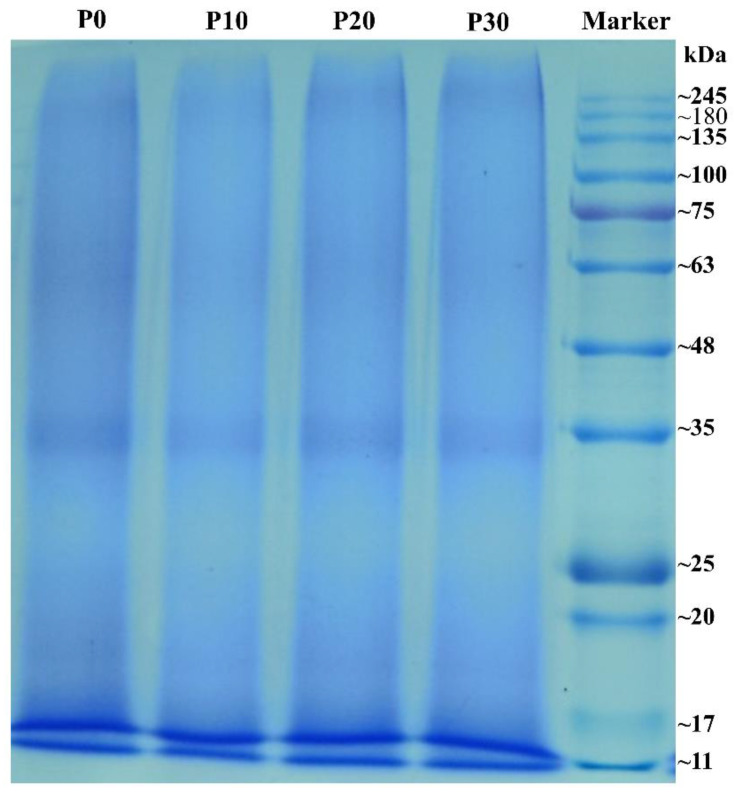
SDS-PAGE analysis of rice protein or homogenized proteins. P0, rice protein; P10, P20 and P30 represented the rice protein blend with homogenization for 0, 10, 20 and 30 min, respectively.

**Figure 2 foods-11-01601-f002:**
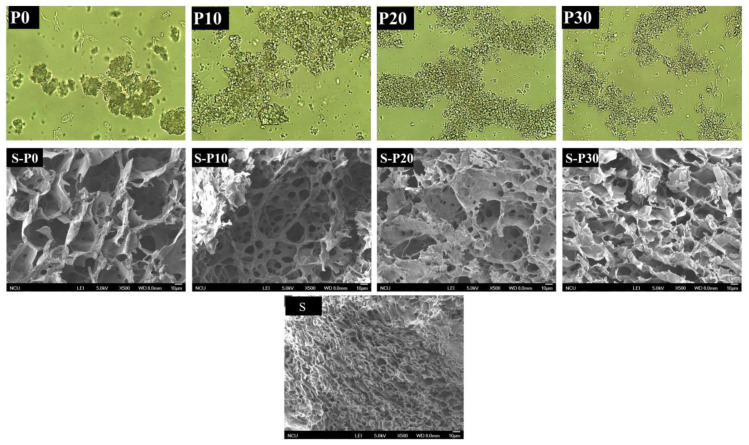
The optical microscope (P0, P10, P20 and P30) and scanning electron microscopy (S, S-P0, S-P10, S-P20 and S-P30) of different samples. P0, rice protein; P10, P20 and P30 represented the rice protein blend with homogenization for 0, 10, 20 and 30 min, respectively. S, rice starch; S-P0, S-P10, S-P20 and S-P30 represented the rice starch blend with rice protein homogenized for 0, 10, 20 and 30 min, respectively.

**Figure 3 foods-11-01601-f003:**
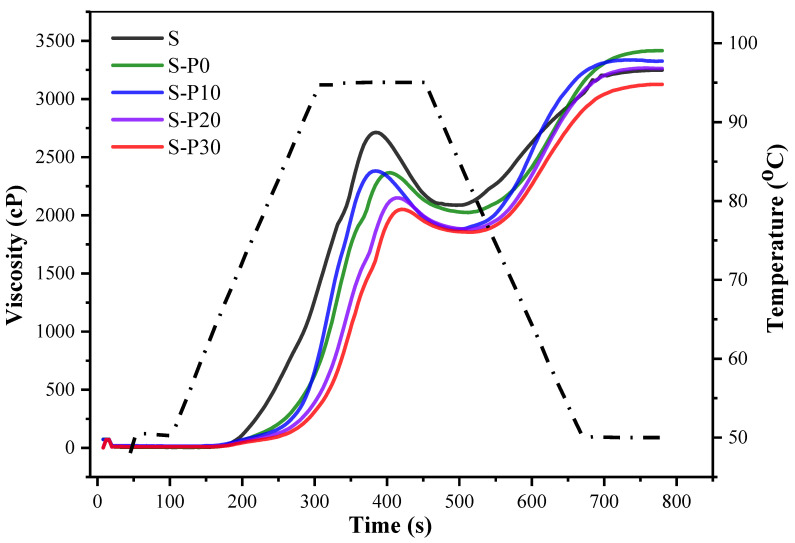
Pasting profiles of rice starch, and rice starch blended with protein or homogenized proteins. S, rice starch; S-P0, S-P10, S-P20 and S-P30 represented the rice starch blend with rice protein homogenized for 0, 10, 20 and 30 min, respectively.

**Figure 4 foods-11-01601-f004:**
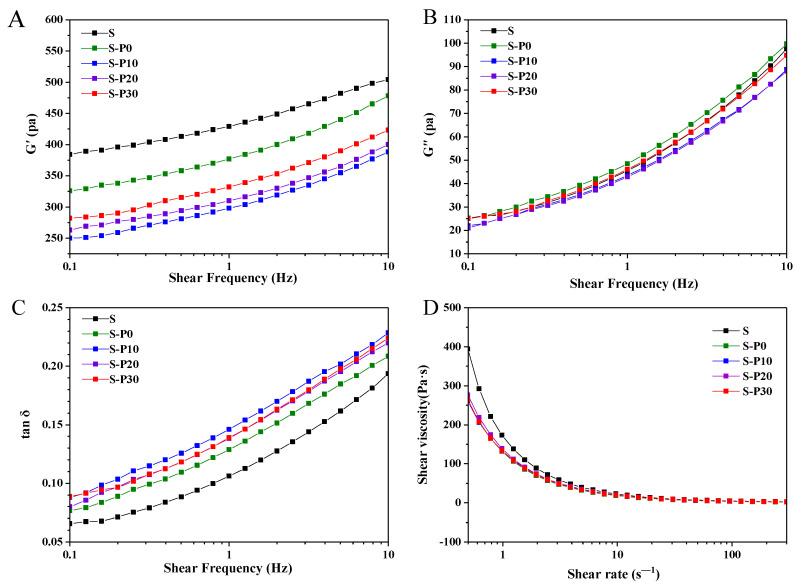
Dynamic rheological properties (**A**–**C**) and flow curves (**D**) of rice starch, and rice starch blended with protein or homogenized proteins. S, rice starch; S-P0, S-P10, S-P20 and S-P30 represented the rice starch blend with rice protein homogenized for 0, 10, 20 and 30 min, respectively.

**Table 1 foods-11-01601-t001:** Characteristics of rice protein or homogenized proteins.

Samples	WHC (g/g)	Particle Size (μm)		
		D_10_	D_50_	D_90_
P0	1.65 ± 0.02 ^d^	4.48 ± 0.01 ^a^	24.2 ± 0.3 ^a^	54.1 ± 0.1 ^a^
P10	1.90 ± 0.03 ^c^	2.27 ± 0.02 ^b^	13.8 ± 0.2 ^b^	34.4 ± 0.4 ^b^
P20	2.06 ± 0.04 ^b^	2.01 ± 0.01 ^c^	10.8 ± 0.1 ^c^	27.4 ± 0.5 ^c^
P30	2.23 ± 0.08 ^a^	1.80 ± 0.01 ^d^	8.74 ± 0.4 ^d^	23.0 ± 0.2 ^d^

Values were mean ± standard deviation of three replicates. Values in the same column followed by different lowercase letters are significantly different at *p* < 0.05 by Tukey’s test. WHC, water holding capacity; P0, rice protein; P10, P20 and P30 represented the rice protein blend with homogenization for 0, 10, 20 and 30 min, respectively.

**Table 2 foods-11-01601-t002:** The pasting properties and flow behavior of rice starch, and rice starch blended with protein or homogenized proteins.

Samples	Pasting Properties						Flow Behavior		
	Peak Viscosity (cP)	Trough Viscosity (cP)	Breakdown Viscosity (cP)	Final Viscosity (cP)	Setback Viscosity (cP)	Pasting Temperature (°C)	K (Pa⋅s^n^)	n	R^2^
S	2718 ± 12 ^a^	2085 ± 10 ^a^	633 ± 22 ^a,b^	3245 ± 4 ^b,c^	1159 ± 13 ^c^	76.4 ± 0.4 ^d^	133.3 ± 1.0 ^a^	0.27 ± 0.02 ^a^	0.92
S-P0	2047 ± 16 ^d^	1426 ± 34 ^d^	712 ± 19 ^a^	3081 ± 47 ^d^	1823 ± 24 ^a^	88.0 ± 0.1 ^c^	117.2 ± 5.7 ^b^	0.23 ± 0.01 ^b^	0.93
S-P10	2156 ± 25 ^c^	1575 ± 66 ^c^	580 ± 45 ^b^	3240 ± 23 ^c^	1717 ± 22 ^a^	90.7 ± 1.3 ^c^	117.4 ± 5.3 ^b^	0.26 ± 0.01 ^a^	0.96
S-P20	2208 ± 12 ^b^	1791 ± 35 ^b^	498 ± 4 ^c^	3313 ± 12 ^a,b^	1491 ± 35 ^b^	91.6 ± 0.5 ^b^	116.5 ± 6.3 ^b^	0.26 ± 0.01 ^a^	0.94
S-P30	2341 ± 23 ^b^	1920 ± 33 ^b^	418 ± 31 ^c^	3327 ± 26 ^a^	1472 ± 3 ^b^	93.1 ± 0.9 ^a^	114.2 ± 6.1 ^b^	0.27 ± 0.01 ^a^	0.95

Values were mean ± standard deviation of three replicates. Values in the same column followed by different lowercase letters are significantly different at *p* < 0.05 by Tukey’s test. S, rice starch; S-P0, S-P10, S-P20 and S-P30 represented the rice starch blend with rice protein homogenized for 0, 10, 20 and 30 min, respectively.

## Data Availability

All data included in this study are available upon request by contact with the corresponding author.

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
