# Peer review of "Effect of Homogenization Modified Rice Protein on the Pasting Properties of Rice Starch"

_foods, 2022, doi:10.3390/foods11111601_

Round 1

Reviewer 1 Report

General overview.

Manuscript foods-1732347 titled “Effect of homogenization modified rice protein on the pasting properties of rice starch”, studied the effects of homogenization on physico-chemical properties of rice proteins, and then to evaluate the potential use of modified rice protein as protein enriching ingredient in starchy food. In general introduction showed easy to read but reported a wrong information and lack a strong background on homogenization effects on proteins and starch in relation to starchy food production. The methods should be improved no statistical analyses was reported. Regarding results and discussions need of same explanation. Are you sure that you found in rice gluten??? In addition, no discussion was reported about the potential use of modified rice protein as protein enriching ingredient in starchy food and the effects on final products. Finally, the conclusion should be enriched.

The real limits of this studies are mainly due to the experimental design, the effects of treatment were tested only on one sample. Are you sure that statistical result (Tukey test) is not influenced by the dataset size? In my opinion strong revision and other data derived from more rice sample are need.

Major comments.

  • Introduction reported that “Native rice protein contains four components: albumin, globulin, gliadin and gluten, among which gluten content is relatively high, accounting for 66%-78% of the total rice protein” are you sure??? Rice is known to be a gluten free cereal. Reported references did not support this sentence indeed Van Der Borght, et al. reported that "apparent MW profiles of rice endosperm proteins allowed classification into six fractions of decreasing apparent MW. Fraction VI contained the low MW albumin, globulin, and prolamin protein material. Fractions IV and V originated from a and b glutelin subunits, respectively. The polypeptides of fraction III consisted of an a and a b subunit linked by an intermolecular disulfide bond. The polypeptides of fractions I and II were dimers, trimers or more highly polymerized forms of the (a–b) glutelin subunit dimer in fraction III. While the work confirmed that rice glutelin is composed of polymers of a and b subunits, remarkably, higher MW glutelin aggregates (fractions I–III) only partly dissociated on reduction". I suppose that gluten is incorrect typing of glutelin. If not, please add more references and explanations.

In addition, more detail about homogenization should be added considering also a reported or hypothetical effect on starch and proteins.

  • The Material and methods sometime appear incomplete. No centesimal composition of sample was reported. In addition, the author evaluated only one sample and it is reductive considering the high variability of proteins, amylopectin and amylose content naturally found in rice. No statistical description was reported. Please add information.
  • The results again show the presence of gluten among rice proteins. I suggest adding scientific evidence of gluten in rice. In addition, if the scope of this work is “to evaluate the potential use of modified rice protein as protein enriching ingredient in starchy food”, more discussion for each paragraph should be added in relation to the effect of treatment on investigated parameters variation and hypothetic consequences in food starch products.
  • I suggest adding a possible application of these treatment in relation to the positive and negative effect on technological properties of starch food derived.

Minor comments.

Line 31-33 Gluten or Glutelin

Line 74 add information about homogenizer (brand, model etc)

Line 92 check if reference in text is correctly reported

Line 119 add information about RVA (brand, model etc)

Line 159 and all text check the number of significant digits reported (e.i. 24.23±0.25 change in 24.2±0.3)

Line 165 add references

Line 182 Gluten or Glutelin

Line 223-225 add references

Line 262 close parentheses

Line 335 check if reference in text is correctly reported

Author Response

We thank the reviewers for their constructive comments and suggestions on our manuscript. We have revised the manuscript in light of these comments and have included a detailed list of responses.  Please kindly see the attachment. 

Reviewer 2 Report

Dear Authors,

The article penned by Wu et al. is interesting and important to the industry. The paper is written in good English, however, there is opportunity for further improvement at times. Authors should proofread their paper before resubmitting process.

Some detailed comments to improve the article

Abstract

Line 15 – please reword the sentence

Introduction

You mention many factors of protein modification, pH as an extremely basic factor is missing, please cite this paper to fill the gap:

Mun et al (2020). Methods to improve rice protein dispersal at moderate pH

Wang et al (2019). Effect of protein aggregates on properties and structure of rice bran protein-based film at different pH

Please also cite the following, it is recent literature and should be included here, in the introduction or discussion of the results

Zhang et al (2019). Effect of rice protein on the water mobility, water migration and microstructure of rice starch during retrogradation

Zhao et al (2020). Comparison of wheat, soybean, rice, and pea protein properties for effective applications in food products

Materials and methods

Line 66 – please  provide detailed information  on rice protein composition, fractions -  you mention that this protein has several fractions, what type of preparation?

Section 2.3.4 what magnitude/magnitudes?

Results and discussion

Line 182 –might be belong to

Lines 223-228 – this section is out of place and should be moved elsewhere

Line 230  - please explain RVA

Line 233 – trough viscosity – NOT THROUGH - please keep an eye on it, because this error (most likely due to the autocorrect ) appears several times in the paper, for example line 251

Table 2 - After all, in what units do the authors measure viscosity? in the table is a different unit, in the text another. Please correct and make conversions, if necessary

Table 2 – please apply the upper index of the letters indicating the statistically significant differences

Line 265 – temperature rose (rise/rose/risen)

Conclusions

very generic, please go into details and articulate and highlight the most important observations,

what applications  in the food industry do you see for the tested systems?

what are the best conditions for the pasting properties?

 I guess all of the comments above can be fixed quickly.

Good luck with corrections:)

Author Response

(The authors gave the same response as above.)

Reviewer 3 Report

Manuscript title:

Effect of homogenization modified rice protein on the pasting 2properties of rice starch

Manuscript ID:  foods-1732347

Authors have tried to look at the impact of homogenization on rice protein. They have looked at how homogenization affect particle size, water holding capacity, viscosity and stability and thereby change physiochemical properties of plant proteins. Manuscript reads well. I have some minor comments.

  1. Typo in line 78 , “mixed the with” remove the .
  2. Correct the symbol for ˚C.
  3. Line 27 for hypoallergenic Cite : A Jain, DM Salunke Proteins: Structure, Function, and Bioinformatics 85 (10), 1820-1830
  4. Font needs correction line 173
  5. Correct the language for line 204, 215-216, 279
  6. In methods sections, line 79, nomenclature used is RS-P0,whereas in the result section it is discussed as S-P0. Please be consistent.
  7. Line 167 Lens culinaris cite V Gaur, V Chanana, A Jain, DM Salunke Acta Crystallographica Section F: Structural Biology and Crystallization 2015 Feb;71(Pt 2):221-5. doi: 10.1107/S2053230X15000734. Epub 2015 Jan 28. PMID: 25664800; PMCID: PMC4321480.
  8. Figure 2 . Use subscript
  9. Figure 3 . What does a dip at 400-500 sec signifies?
  10. Figure 4. panel A , Explain why G’ of SP30 is in between SP10& SP20. Please elaborate.
  11. Correct figure information. Fig 5C, 5D.
  12. In general, there is a need of English grammar correction.

Author Response

(The authors gave the same response as above.)

Round 2

Reviewer 1 Report

 Thank  to the authors for appreciating all suggestions. However I have onother request if possible to   check the significant digits numbers in table and text. (ie 24.23+- 0.25 should be apporximate to  24.2 +- 0.3)

Author Response

 Thank  to the authors for appreciating all suggestions. However I have on other request if possible to  check the significant digits numbers in table and text. (ie 24.23+- 0.25 should be apporximate to  24.2 +- 0.3)

Response: We thank the reviewers for their constructive comments and suggestions on our manuscript. The significant digits in table and text was modified as good suggestion. When the values was higher than 10, the value was kept a decimal (such as 24.2±0.3 um).  When the values was lower than 10, the value was kept two decimal (such as 8.74±0.4 um).